# A Polynomial-Exponent Model for Calibrating the Frequency Response of Photoluminescence-Based Sensors

**DOI:** 10.3390/s20164635

**Published:** 2020-08-18

**Authors:** Angel de la Torre, Santiago Medina-Rodríguez, Jose C. Segura, Jorge F. Fernández-Sánchez

**Affiliations:** 1Department Signal Theory, Networking and Communications, University of Granada, 18071 Granada, Spain; atv@ugr.es (A.d.l.T.); segura@ugr.es (J.C.S.); 2By Techdesign S.L., 28500 Arganda del Rey (Madrid), Spain; santiago.medina@by.com.es; 3Department of Analytical Chemistry, University of Granada, 18071 Granada, Spain

**Keywords:** calibration, chemical sensor, photoluminescence, oxygen sensing, frequency response, Stern–Volmer model, Lehrer model, Demas model, polynomial-exponent model

## Abstract

In this work, we propose a new model describing the relationship between the analyte concentration and the instrument response in photoluminescence sensors excited with modulated light sources. The concentration is modeled as a polynomial function of the analytical signal corrected with an exponent, and therefore the model is referred to as a polynomial-exponent (PE) model. The proposed approach is motivated by the limitations of the classical models for describing the frequency response of the luminescence sensors excited with a modulated light source, and can be considered as an extension of the Stern–Volmer model. We compare the calibration provided by the proposed PE-model with that provided by the classical Stern–Volmer, Lehrer, and Demas models. Compared with the classical models, for a similar complexity (i.e., with the same number of parameters to be fitted), the PE-model improves the trade-off between the accuracy and the complexity. The utility of the proposed model is supported with experiments involving two oxygen-sensitive photoluminescence sensors in instruments based on sinusoidally modulated light sources, using four different analytical signals (phase-shift, amplitude, and the corresponding lifetimes estimated from them).

## 1. Introduction

An accurate operation of photoluminescence sensors requires the appropriate calibration of the models describing the relationship between the analytical signal and the concentration of the analyte to be determined [1,2]. The calibration consists in the estimation of the model parameters that fit some calibration data according to some optimization criterion. Therefore, given a model (with *M* parameters) and some calibration data, the calibration procedure can be mathematically formulated as the multidimensional search of the *M* parameters minimizing the error between the calibration data and the model [3].

The classical Stern–Volmer model [1] provides a basic description of the photoluminescence phenomenon. This simple model (with two parameters) describes the luminescent response with a first order differential equation. Even though the Stern–Volmer model provides a qualitative description valid for a wide range of photoluminescence sensors [1,4], this calibration model is insufficient for an accurate description of most of the luminescence instruments because of the heterogeneous distribution of the analyte, illumination and luminescence in the sensing phase [3,5,6,7,8].

In order to provide a more accurate description of the relationship between the analyte concentration and the instrument response, when the 1-site Stern–Volmer model is insufficient, the 2-site Lehrer [9] and Demas [6,7,10,11] models (which are extensions of the former) are commonly used for calibration. Additionally, the Demas model provides scalability, and can be extended to more than 2 sites [11,12]. More complex models have also been proposed in the literature, in order to accurately describe the interaction between the quencher and the luminophore in the sensing phase, for example, by modeling the diffusion of the quencher in the sensor volume [8,13,14], the spatial distribution of the luminescence lifetime [12,15,16], or the dynamics of the luminescence [17]. Such models involve complex mathematical description (including integrals for the volume of the luminescent material) and a large number of parameters to be fitted. Even though these models are useful to understand specific aspects of the quenched photoluminescence, their applicability for calibrating instruments is limited due to the number of parameters to be fitted [18].

The model complexity is a relevant aspect to be considered for calibrating an instrument. A model with more parameters to be fitted provides a more accurate fitting of the calibration data. However, the calibration data are always affected by noise, and if the number of parameters is too large or the calibration data are insufficient (in terms of statistical robustness), the calibration curve will be affected by over-fitting (low calibration error but large instrumental error because the calibration curve is over-adapted to the errors in the calibration data) [19,20]. Additionally, fitting a complex model implies a multidimensional search of the *M* parameters of the calibration curve, and the search and convergence becomes more difficult as *M* increases [21]. In that sense, a good calibration model is one providing a good accuracy with a low number of parameters [2].

In this work, we propose a new model to describe the relationship between the analytical response and the analyte concentration in chemical sensors based on photoluminescence. In its basic form, the proposed model can be considered an extension of the 2-parameter Stern–Volmer model, by including an exponent to the analytical signal as an additional parameter to be fitted (thus yielding to a 3-parameter model). The proposed model can be scaled and admits a simple extension to *M* parameters (with *M* arbitrarily large), which provides flexibility of the model depending on the availability of calibration data. For *M* parameters, the model is defined with a M−2 degree polynomial and an exponent (its calibration requires M−1 coefficients for the polynomial and an additional parameter for the exponent), and therefore the proposed model is referred to as the polynomial-exponent model.

The proposed model is motivated by the limitations of the classical models, particularly for the description of photoluminescence sensors using modulated light sources [22,23]. The polynomial-exponent model is compared in this paper with the classical models commonly used for calibrating photoluminescence sensors: the Stern–Volmer, the Lehrer, and the 2-site Demas models (requiring 2, 3, and 4 parameters, respectively). The proposed and the classical models have been evaluated in experiments involving two oxygen sensing phases in two different photoluminescence measuring systems based on sinusoidally modulated light sources. The experimental results show that the proposed polynomial-exponent model provides a good trade-off between accuracy and number of parameters. It is applicable to the amplitude, phase-shift, or lifetime measurements of luminescence, and it is thought to be of potential utility for enzymatic biosensors coupled with the optical oxygen transducer, and related chemical sensors employing quenching/enhancement of luminescence.

## 2. Limitations of the Classical Calibration Models

The polynomial-exponent model that we propose in this manuscript is motivated by the limitations of the classical models conventionally considered for calibrating photoluminescence sensors when they are applied to measurements in the frequency domain. In this section, we analyze in detail the classical calibration models and their limitations when applied in phase-resolved photoluminescence.

### 2.1. Photoluminescence Response with Modulated Excitation

#### 2.1.1. The Stern–Volmer Equation

Luminescence is the result of a radiative deactivation of the molecules of a luminescent material from an excited state to the fundamental state. The simplest first order linear photoluminescence model describes the population of the excited state N(t) linearly increasing with the illumination $x_{exc}\left(t\right)$ and linearly decreasing according to the radiative and non-radiative deactivation constants $\Gamma_{r}$ and $\Gamma_{nr}$ [1]:(1)$$\frac{d\;N(t)}{d\;t}=a_{0}x_{exc}\left(t\right)-\left(\Gamma_{r}+\Gamma_{nr}\right)N\left(t\right),$$
where *t* is the time variable and $a_{0}$ is a constant that relates the illumination used as excitation with the increase of the excited state population in the luminescent material. This is a non-homogeneous first order linear differential equation characterized by a lifetime $\tau_{q}$ inversely proportional to the sum of the deactivation constants. In the presence of a quencher, the simplest quenching model suggests a linear increase of the non-radiative deactivation constant with the quencher concentration *C*, thus yielding to the Stern–Volmer equation:(2)$$\tau_{q}=\frac{1}{\Gamma_{r}+\Gamma_{nr}}=\frac{1/\Gamma_{r}}{1+\Gamma_{nr}/\Gamma_{r}}=\frac{\tau_{0}}{1+kC}$$
where $\tau_{0}$ is the inverse of $\Gamma_{r}$ (i.e., the lifetime at null quencher concentration) and *k* is the Stern–Volmer constant [1,4,24]. The photoluminescent emission $x_{em}\left(t\right)$ is proportional to the radiative decay, i.e., to the product $\Gamma_{r}\cdot N\left(t\right)$, and therefore the photoluminescence is described by a similar differential equation:(3)$$\frac{d\;x_{em}\left(t\right)}{d\;t}=a_{1}x_{exc}\left(t\right)-\frac{x_{em}\left(t\right)}{\tau_{q}}$$
where the coefficient $a_{1}$ provides the relation between excitation and emission.

#### 2.1.2. Frequency Response of the Stern–Volmer Model

In the case of an illumination source modulated with a sinusoidal or periodic signal, the response of the photoluminescent sensor is better described through its frequency response. The differential equation is transformed to the frequency domain [25,26,27] by substitution of the derivative operator by jω (where *j* is the imaginary unity, ω=2πf is the angular frequency, in radians per second, and *f* is the frequency in Hz):(4)$$j\omega X_{em}\left(j\omega \right)=a_{1}X_{exc}\left(j\omega \right)-\frac{1}{\tau_{q}}X_{em}\left(j\omega \right)$$
and the frequency response can be obtained from this equation:(5)$$H\left(j\omega \right)=\frac{X_{em}\left(j\omega \right)}{X_{exc}\left(j\omega \right)}=\frac{\tau_{q}a_{1}}{1+j\omega \tau_{q}}$$
which is usually rewritten as:(6)$$H\left(j\omega \right)=M_{0}\frac{\tau_{q}}{\tau_{0}}\frac{1}{1+j\omega \tau_{q}}$$
where $M_{0}$ is the modulation-factor at null quencher concentration and low frequency (i.e., the ratio between the emission and excitation at these conditions). In phase-resolved luminescence spectroscopy, the frequency response describes the relationship between the excitation and the emission for each frequency component in the excitation signal. At each frequency, H(jω) is a complex number, with modulus and argument, the modulus representing the ratio between the amplitudes of the emission and excitation sinusoidal components of frequency ω, and the argument representing the phase-shift between both sinusoidal components [27,28].

According to this linear first order model, the modulation-factor predicted for a given concentration *C* and modulation frequency ω is [1,22]:(7)$$m\left(C,j\omega \right)=\sqrt{{\left|H\left(j\omega \right)\right|}^{2}}=M_{0}\frac{\tau_{q}\left(C\right)}{\tau_{0}}\frac{1}{\sqrt{1+{\left(\omega \tau_{q}\left(C\right)\right)}^{2}}}$$
and the phase-shift is [1,22]:(8)$$\phi \left(C,j\omega \right)=-\arctan(\omega \tau_{q}\left(C\right))$$

#### 2.1.3. Analyte Determination with Modulated Illumination Sources

From the previous equations, the lifetime $\tau_{q}$ can be estimated either from the modulation-factor or from the phase-shift [22,23]:(9)$$\tau_{\phi }\left(C\right)=\frac{-\tan(\phi (C,j\omega ))}{\omega }\;\;\tau_{m}\left(C\right)=\tau_{0}\frac{m\left(C,j\omega \right)/M_{0}}{\sqrt{1-{\left(\omega \tau_{0}\right)}^{2}{\left(m\left(C,j\omega \right)/M_{0}\right)}^{2}}}$$

However, in phase-resolved spectroscopy, the modulation-factor at null concentration is difficult to be measured at null frequency, while it is easier to be estimated at the modulation frequency. The modulation-factor at the modulation frequency ω for null concentration is given by:(10)$$m_{0}\left(j\omega \right)=m\left(0,j\omega \right)=\frac{M_{0}}{\sqrt{1+{\left(\omega \tau_{0}\right)}^{2}}}$$
and therefore the following expression is more practical for the estimation of the lifetime based on the modulation-factor:(11)$$\tau_{m}\left(C\right)=\tau_{0}\frac{m\left(C,j\omega \right)/m_{0}\left(j\omega \right)}{\sqrt{1+{\left(\omega \tau_{0}\right)}^{2}\left[1-{\left(m\left(C,j\omega \right)/m_{0}\left(j\omega \right)\right)}^{2}\right]}}$$

(this expression requires the parameter $\tau_{0}$, usually estimated as $\tau_{\phi }$ at null concentration).

In summary, a phase-resolved photoluminescence sensor (i.e., using a modulated light source as excitation), described by a first order linear model, can easily be calibrated with Equations (Equation 9), (Equation 11) and (Equation 2). At a given frequency ω and concentration *C*, the modulation-factor m(C,jω) and the phase shift ϕ(C,jω) can be measured from an analysis of the excitation and emission signals $x_{exc}\left(t\right)$ and $x_{em}\left(t\right)$. From them, the modulation-factor and the phase-shift based lifetimes $\tau_{m}\left(C\right)$ and $\tau_{\phi }\left(C\right)$ are estimated with Equations (Equation 9) and (Equation 11). By collecting measurements at different concentrations, the Stern–Volmer model can be calibrated using Equation (Equation 2), i.e., the parameters of the Stern–Volmer model ($\tau_{0}$ and *k*) providing the best fitting of the calibration data set can be estimated. After the calibration, from a new measurement (with an excitation signal modulated at frequency ω) obtained at an unknown concentration, the modulation-factor and phase-shift can be measured, the corresponding lifetimes can be derived (either $\tau_{m}$ or $\tau_{\phi }$), and using the calibrated Stern–Volmer model, the concentration can be estimated, providing the analyte determination $C(\tau_{q})$.

As can be noted from the equations, for the first order model, the luminescence lifetime only depends on the concentration, and the lifetime estimations obtained either from phase-shift or modulation-factor and at different frequencies should be identical. Due to noise (mainly that affecting the emission signal) and the error propagation, the accuracy of a phase-resolved photoluminescence sensor strongly depends on the modulation frequency, and multifrequency sensors with an appropriate combination of $\tau_{m}$ and $\tau_{\phi }$ estimated at several frequencies provide the best results for the complete range of concentrations [22,23,29,30].

### 2.2. Limitations of the Stern–Volmer Model

The Stern–Volmer model provides a very good qualitative description of a wide range of photoluminescent sensors. The accuracy provided by this model is acceptable for some sensing phases, but it is insufficient for most of them, due to deviations of the photoluminescence from the first order linear model [1,3,4,11]. These deviations are associated with the presence of different excitation and/or deactivation mechanisms, the distribution of the quencher in the volume of the luminescent material, biased measurements of the modulation-factor or phase-shift due to instrumental configuration, etc. Taking into account the mathematical formulation of the model, two different situations should be considered: (a) the relationship between the quencher concentration and the non-radiative deactivation constant is not linear (i.e., Equation (Equation 2) is insufficient); or (b) the order of the differential equation is higher than one.

From the mathematical perspective, both situations are completely different, since in the first case, the photoluminescence sensor is a first order system (i.e., with a monoexponential evolution of the luminescent response, with a well defined lifetime, and providing identical lifetime estimations from modulation-factor or from phase-shift and at whatever modulation frequency), while in the second case there is not a well-defined luminescence lifetime, the estimated values depend on the estimation procedure (so they are referred to as “apparent lifetimes”) and therefore the calibration parameters depend on the sensor configuration (modulation frequency, whether the apparent lifetime is estimated either from the modulation-factor or from the phase-shift, etc.).

For the first situation (first order differential equation but nonlinear relationship between quencher concentration and deactivation constant), the Lehrer model is an extension of the Stern–Volmer model [1,9]:(12)$$\tau_{q}=\tau_{0}\left(\frac{x}{1+kC}+\left(1-x\right)\right)$$
where a proportion *x* of the luminescent material is assumed to be affected by the quencher while the rest, 1−x, is not affected. Compared with the Stern–Volmer model, the 3-parameter Lehrer model increases the flexibility in the description of the $\tau_{q}\left(C\right)$ curves, providing a better calibration for many photoluminescence sensors.

The Demas model can also be considered an extension of the Stern–Volmer model [1,7]:(13)$$\tau_{q}=\tau_{0}\left(\frac{x}{1+k_{1}C}+\frac{\left(1-x\right)}{1+k_{2}C}\right)$$
and represents two independent quenching contributions, each one with a different Stern–Volmer constant. The 4-parameter Demas model increases the flexibility with respect to the Stern–Volmer and Lehrer models and provides more accurate calibrations. Additionally, the Demas model is scalable (and can be extended to more than two sites by including more terms):(14)$$\tau_{q}=\sum_{n}\frac{\tau_{0,n}}{1+k_{n}C}$$
where $\tau_{0,n}$ is the contribution of the *n*-th site to the lifetime at null quencher concentration.

For the second situation (higher order differential equation), a multi-exponential photoluminescence model is proposed [1,22,23], providing the following frequency response:(15)$$H\left(C,j\omega \right)=\sum_{n}M_{0,n}\frac{\tau_{q,n}}{\tau_{0,n}}\frac{1}{1+j\omega \tau_{q,n}}$$
where the index *n* represents each mono-exponential process contributing to the multi-exponential response; for each process *n*, $M_{0,n}$ is the modulation factor at low frequency and null concentration, $\tau_{0,n}$ is the lifetime at null concentration, and the relation between the concentration and the lifetime for each mono-exponential process is described by a Stern–Volmer equation [1,24]:(16)$$\tau_{q,n}\left(C\right)=\frac{\tau_{0,n}}{1+k_{n}C}$$
with its corresponding Stern–Volmer constant $k_{n}$. In this model, each mono-exponential process requires three parameters to be calibrated ($M_{0,n}$, $\tau_{0,n}$, and $k_{n}$). By transforming the frequency response into the corresponding differential equation, it can be demonstrated that an *N*-process model is equivalent to a *N*-order differential equation.

Interestingly, the Demas multi-site first order model and the multi-exponential model provide an equivalent equation for low frequencies. Effectively, if $\omega \tau_{q,n}\ll 1$ for all the processes, the dependence with frequency disappears, and Equation (Equation 15) can be rewritten as:(17)$$H\left(C\right)=\sum_{n}\frac{M_{0,n}}{1+k_{n}C}$$
and, since the modulation-factor is proportional to the lifetime at low frequency (according to Equation (Equation 11)), both models are identical at low frequency.

However, the differences between both models become significant for phase-resolved photoluminescence sensors using higher modulation frequencies (or for multifrequency photoluminescence). Experimental results suggest that deviation from the simple Stern–Volmer model is usually better explained with a multiexponential model than with a multi-site quenching of the lifetime, since the apparent lifetimes estimated at different frequencies, either from modulation-factor or phase-shift, provide inconsistent estimations [23,29,30,31,32], while a multiexponential model provides a globally consistent description of the photoluminescence response at different modulation frequencies [23].

Even though the multiexponential model is preferable for a global description of the photoluminescence multifrequency response, the sensor calibration is usually performed with the Demas model and using apparent lifetimes (either from modulation-factor or phase-shift based and estimated at a single frequency) because the combination of the different measured parameters in a single analyte determination is not very common and it is a relatively recent proposal [23,29,30]. In addition, the Demas model is expected to provide a reasonable calibration when it is applied to the apparent lifetimes because at moderate frequencies the relation between m(C,jω) and $\tau_{m}\left(C\right)$, or between ϕ(C,jω) and $\tau_{\phi }\left(C\right)$, is close to linear, and usually very high modulation frequencies are not applied in phase-resolve photoluminescence (because the emission amplitude is low at high modulation frequencies, and the response would be too noisy for an accurate analyte determination) [23].

### 2.3. Limitations of the Demas Model

According to the previous discussion, even though the Demas multi-site description is not appropriate for a multiexponential photoluminescence sensor, at very low frequencies, it is perfectly consistent (since it is mathematically equivalent to the multiexponential model). On the other hand, if the deviation from the Stern–Volmer model is moderate, or at moderate frequencies (relative to the involved lifetimes), the Demas model provides a reasonable calibration, usually with accuracy significantly improved with respect to the Stern–Volmer model (because the inconsistency can be compensated by the model flexibility). However, the inconsistency of the Demas model limits its accuracy in multiexponential photoluminescence sensors excited with modulated light sources.

Additionally, the instrumental bias (affecting the modulation-factor and the phase-shift measurements) and its propagation to the apparent lifetimes also contribute to limiting the accuracy of the Demas-based calibration. An accurate estimation of the modulation-factor and phase-shift requires the acquisition of the excitation and emission signals $x_{exc}\left(t\right)$ and $x_{em}\left(t\right)$ and the subsequent digital signal processing [1,27,28]. The illumination is usually provided by an electrical signal which feeds a driver circuit connected to the illumination source, and the excitation signal is acquired by sampling the electrical signal used for the illumination system. The acquisition of the excitation signal is not difficult since this signal is usually provided by a signal generator, with large amplitude and high signal-to-noise ratio. However, the acquisition of the emission signal in a photoluminescence sensor is usually affected by several problems. The photoluminescence response is converted into an electrical signal by a photodetector (for example, a photodiode, a phototransistor, or a photomultiplier tube), and the electrical signal has to be amplified and sampled. Depending on the intensity of the photoluminescence response and the required sensitivity, large gains are usually required in the transduction, and usually the transductors and the amplification circuitry cause linear and nonlinear distortions (group delay, frequency response associated with the instrument including gain and phase-shift distortions, nonlinear distortion due to the transducer response, etc.). A significant bias in the estimation of the phase-shift is very common in photoluminescence sensors (associated with group delay in the photodetector and frequency response of the photodetector and preamplifier). The main source of bias in the amplitude measurements would be associated with the polarization of the electronic and optoelectronic components, but in a linear system it does not affect frequency based measurements (only at null frequency). Therefore, the bias in the modulation-factor is expected to be moderate and would be associated with nonlinear effects in the photodetector.

These effects can be estimated, calibrated, and compensated in order to remove them from the frequency response of the photoluminescence sensor and therefore to obtain a more consistent calibration of the sensor using a Demas or a multiexponential model. However, an accurate calibration and compensation of the instrumentation effects for the operation conditions (or a join calibration of both the instrumentation and the photoluminescent subsystems) are not easy [1,24,27,28].

## 3. The Polynomial-Exponent Model

The calibration based on the Demas model presents two advantages: firstly, the model is based on the physics of the sensing phase and therefore it provides a physical interpretation of the calibration parameters. On the other hand, it is a scalable model. However, it also presents some disadvantages: firstly, the consistency of the model is limited for sensors that deviate from the Stern–Volmer model operating at moderate frequency. Secondly, an instrumental bias would increase the inconsistency of the model. Finally, the calibration of a *N*-sites model (requiring the optimization of 2N parameters) can be simplified to a *N*-dimensional search [2], but it becomes unpractical for large *N*, thus limiting the scalability of the model.

Taking into account the advantages of the Demas-based calibration, and also its limitations (associated with inconsistencies when the luminescent sensing phases requires a differential equation of order higher than one, or because of instrumental nonlinearity or bias), we propose to describe the relationship between the analytical signal ϕ (where ϕ would be either the phase-shift, the modulation-factor or the corresponding lifetimes) and the concentration C(ϕ) with a polynomial function applied to $\phi^{\alpha }$, i.e., applied to the analytical signal corrected with an exponent:(18)$$C\left(\phi \right)=\sum_{n=0}^{p}a_{n}\phi^{n\alpha }=a_{0}+a_{1}\phi^{\alpha }+a_{2}\phi^{2\alpha }+\ldots +a_{p}\phi^{p\alpha }$$

The calibration of this polynomial-exponent model (which will be referred to as PE model) requires the estimation of p+2 parameters, where *p* is the polynomial degree (p+1 parameters for the polynomial coefficients and an additional one for the exponent α).

Even though the polynomial-exponent model is not intended for a physical description of the photoluminescence (but for the calibration of photoluminescence sensors), it provides a very simple estimation of the sensor response at null concentration $\phi_{0}$ and the sensitivity at a given conentration, $K_{C}$. These parameters are defined as:(19)$$\phi_{0}\equiv \lim_{C\to 0}\phi \;K_{C}\equiv \frac{1}{\phi (C)}\left|\frac{\partial \phi (C)}{\partial C}\right|\;K_{0}\equiv \lim_{C\to 0}\frac{1}{\phi (C)}\left|\frac{\partial \phi (C)}{\partial C}\right|$$
and, for the PE model, they are obtained, respectively, from the polynomial roots and from the first derivative of the polynomial. For example, in the case of the first degree PE model (PE1), the parameters providing the physical description of the model are:(20)$$\phi_{0}={\left(\frac{-a_{0}}{a_{1}}\right)}^{1/\alpha }\;K_{0}=\left|\frac{1}{a_{0}\alpha }\right|\;K_{C}=\left|\frac{1}{\alpha a_{1}\phi_{C}^{\alpha }}\right|$$
and those for the PE2 model (second degree) are:(21)$$\phi_{0}={\left(\frac{-a_{1}+\sqrt{a_{1}^{2}-4a_{0}a_{2}}}{2a_{2}}\right)}^{1/\alpha }\;K_{0}=\left|\frac{1}{\alpha a_{1}\phi_{0}^{\alpha }+2\alpha a_{2}\phi_{0}^{2\alpha }}\right|\;K_{C}=\left|\frac{1}{\alpha a_{1}\phi_{C}^{\alpha }+2\alpha a_{2}\phi_{C}^{2\alpha }}\right|$$

($\phi_{C}$ being the analytical signal at the concentration *C*).

In fact, if the analytical signal is the lifetime, the first degree PE model can be considered as an extension of the Stern–Volmer model:(22)$$C\left(\tau \right)=a_{0}+a_{1}\tau^{\alpha }\;\;\tau \left(C\right)={\left(\frac{-a_{1}/a_{0}}{1-C/a_{0}}\right)}^{-1/\alpha }\;\;\tau_{SV}\left(C\right)=\frac{\tau_{0}}{1+kC}$$
and the comparison of both expressions of the lifetime reveals that the Stern–Volmer model is a particular case of the PE1 model for α=−1, where the lifetime at null concentration is $\tau_{0}=-a_{1}/a_{0}$ and the sensitivity (or Stern–Volmer constant) is $K=-1/a_{0}$.

The scalability of the PE model is easier than in the Demas model because the estimation of the polynomial coefficients for a given exponent α does not require any search (but a one-step matrix inversion), and the optimization of the *p*-degree PE-model requires a one-dimensional search (for α) independently of the polynomial degree (details about the PE-model calibration are provided in Appendix A).

In this work, we evaluate 1st and 2nd degree PE models (PE1 and PE2) requiring the estimation of three and four parameters, respectively. Additionally, in order to analyze the role of the exponent included in the model, we also evaluate the PE2 model with the exponent constrained to α=−1, i.e., a 3 parameter model which will be referred to as P2 model (note that P1 model corresponds to the Stern–Volmer model, and, therefore, the P2 model could be considered another three-parameter extension of the Stern–Volmer model). Octave/MatLab functions for the calibration of the P2, PE1, and PE2 models are provided in Appendix A. We also evaluate the Stern–Volmer, Lehrer, and Demas models as reference. These reference models, require the estimation of 2, 3, and 4 parameters, respectively. In these models, the response at null concentration, $\phi_{0}$, is a parameter of the model, and the sensitivity at *C*=0 is $K_{0}=k$, $K_{0}=x\cdot k$ and $K_{0}=x\cdot k_{1}+\left(1-x\right)\cdot k_{2}$ for the Stern–Volmer, Lehrer, and Demas models, respectively.

## 4. Experimental Design

The accuracy of the proposed PE model for calibrating photoluminescence-based sensors has been evaluated with experiments involving two different sensing phases for oxygen detection. In these experiments, the phase-shift or the modulation-factor were measured by phase-resolved luminescence spectroscopy in the frequency domain.

The first sensing phase, Platinum(II) 5,10,15,20-meso-tetrakis-(2,3,4,5,6-pentafluorophenyl)- porphyrin immobilized in polystyrene (PtTFPP/PS), is an oxygen-sensing film coated at the end of an optical fibre. The photoluminescence sensor was designed for measuring oxygen in gas phase ($pO_{2}$) in the range 0.5–20 kPa. The sensing film was illuminated with an ultraviolet LED excited with a sinusoidal signal at a frequency of 5145 Hz. The photoluminescent emission was transduced and amplified with a photomultiplier tube and a low-noise preamplifier. Both the excitation and emission electrical signals were digitized with an oscilloscope, and digital signal processing implementing the I/Q method was applied to obtain the modulation-factor and the phase-shift for each measurement. Details about this sensing film, the instruments involved in the acquisition of the excitation and emission signals, and the I/Q method for estimating the modulation-factor and phase-shift from the recorded signals can be found in our previous works [23,31].

The second sensing phase (N1008-AP200/19) is an iridium-based oxygen-sensing dye, Ir(2-(2,4-difluorophenyl) pyridine) 2(4,7-diphenyl-1, 10-phenanthroline), immobilized into AP200/19, a nanostructured aluminium oxide-hydroxide solid support. The sensor using this sensing-phase was designed for measuring oxygen in gas phase at low partial pressures (in the range 0.25–10 kPa). The sensing film was illuminated with an ultraviolet LED excited with a sinusoidal signal at a frequency of 30,100 Hz. The photoluminescent emission was transduced and amplified with a photomultiplier tube and a low-noise preamplifier. Both the excitation and emission electrical signals were processed with a dual-phase lock-in amplifier (LIA) (SR830, Stanford Research Systems, Sunnyvale, CA, USA). The LIA instrument provided the phase-shift between the excitation and emission signals. Details about the N1008-AP200/19 sensing film, the involved instruments, and the procedure for estimating the phase-shift can be found in our previous works [33,34].

Six different calibration experiments were defined with the measurements obtained with these sensors: experiments 1, 2, 3, and 4 involve the PtTFPP/PS sensor, and provide the concentration ($O_{2}$ partial pressure, kPa) using the phase-shift ϕ, the phase-shift based lifetime $\tau_{\phi }$, the modulation-factor *m*, or the modulation-factor based lifetime $\tau_{m}$ as analytical signal, respectively. Experiments 5 and 6 involve the N1008-AP200/19 sensor and provide the $O_{2}$ partial pressure using the phase-shift ϕ and the corresponding lifetime $\tau_{\phi }$, respectively. Obviously, the pair of calibration experiments involving a measurement (either ϕ or *m*) and the corresponding lifetime are correlated, since they are based in the same primary measurements.

For the experiments 1, 2, 3, and 4, the analytical signal was obtained at different oxygen concentrations, with 250 measurements registered at each concentration. For the experiments 5 and 6, 100 measurements are available at each concentration. The data acquisition was performed with increasing oxygen concentration and leaving a sufficient time at each new concentration in order to allow the stabilization of the gas station and the sensing phase before the data recording. For each experiment, the measurements were randomly assigned to disjoint calibration and evaluation subsets (125 measurements at each concentration for the experiments 1, 2, 3, and 4; fifty measurements at each concentration for the experiments 5 and 6). The data in the calibration datasets were averaged for the calibration procedures (in order to accurately analyze the residual of the fitting curves). The data in the evaluation datasets were used individually in order to evaluate the impact of the calibration procedures over the accuracy in the oxygen determination.

For each experiment, tables with the mean and standard deviations of the analytical signals at each concentration (for the calibration and evaluation datasets) are provided in Appendix A.

## 5. Results and Discussion

The calibration dataset for each experiment was used for calibrating the Stern–Volmer (SV), Lehrer (L), Demas (D), 2nd degree polynomial (P2), and 1st and 2nd degree polynomial-exponent (PE1 and PE2) models. The optimization criterion applied for calibration was the minimization of the relative error in the concentration (in this case $pO_{2}$) [2]. The SV and P2 optimizations are obtained directly from the calibration dataset (these models do not require any iterative procedure for the calibration). Since the L, PE1, and PE2 optimizations require a one-dimensional search, the calibration was configured for 100 iterations for these models. The D optimization was performed with 500 iterations (since it involves a two-dimensional search). The SV, L, and D models have been calibrated with the Octave/MatLab functions provided in our previous work [2]. The P2, PE1, and PE2 models have been calibrated with the Octave/MatLab functions provided in Appendix A.

### 5.1. Calibration Curves

Figure 1 represents the average analytical signals (phase-shift, modulation-factor or the corresponding lifetimes) obtained at each concentration in each experiment (black circles: mean values of the calibration data; black error lines: 95% confidence interval of the calibration data) and the calibration curves provided by the SV, L, D, P2, PE1, and PE2 models. All the models provide an appropriate fitting of the calibration datasets with the exception of the SV, which shows a moderate deviation with respect to experimental data in experiments with the PtTFPP/PS sensor, and an unacceptable deviation in the case of the N1008-AP200/19 sensor. The differences among the other models are difficult to be appreciated in these curves.

Some detailed results of the calibration procedures for each experiment and calibration model are provided in Appendix A. The complementary information includes: the model parameters (estimations and standard errors), the determination coefficient $R^{2}$, the calibration error minimized in the calibration procedure, the model response at null concentration, the sensitivity at 0 kPa, 1 kPa, and 10 kPa, and figures with the calibration curves and the deviation of the obtained calibration curves relative to the reference concentrations in the calibration datasets.

### 5.2. Error of the Calibration Models

The performance of the calibration models was evaluated in terms of the RMS relative error (root mean square error relative to the measured $pO_{2}$) expressed in percentage. Table 1 presents the RMS relative error provided by each calibration model in each experiment. The number of parameters to be fitted in each calibration model is also indicated (in parentheses) for a consistent comparison.

The table includes (A) results for the average values in the calibration datasets (providing an evaluation of the calibration accuracy) and also (B) results for the individual values in the evaluation datasets (providing an estimation of the error in the oxygen determination using each calibration curve). The last column represents the RMS relative error averaged for the six experiments. Appendix A includes figures with the relative error (estimated for the evaluation datasets) as a function of the analyte concentration, for each experiment and calibration model.

These results show that the SV model is insufficient for describing the photoluminescence response, particularly for the iridium-based sensor (experiments 5 and 6). The three-parameter models improve the 2-parameter SV model in all the experiments. Similarly, the 4-parameter models improve the 3-parameter models in most cases, with the exception of experiments 3 and 6, where the PE1 model outperforms the D model.

The proposed PE model provides a very good calibration for all the experiments. In the case of 3-parameter models, the PE1 model provides the best results in all the cases, with the exception of the calibration of experiment 4 (with results slightly worse than those with the P2 and L models). In the case of the 4-parameter models, the PE2 is clearly better in most of the experiments (only for experiment 4 the result of PE2 model is slightly better than that of the D model). The average results also suggest a more appropriate calibration provided by the proposed PE model (with both the calibration and the evaluation datasets).

The improvement of the PE1 and PE2 models with respect to the classical L and D models is more significant for the N1008-AP200/19 sensor (experiments 5 and 6). This is probably associated with the fact that this sensor departs too much from the Stern–Volmer description (suggesting important limitations of a description of the photoluminescence system with a first order differential equation), and then the effect of the L or D model inconsistencies becomes more relevant for this sensor. The strong deviation from the first order model observed in experiments 5 and 6 could be associated with the nanostructured support used in the N1008-AP200/19 sensor. On the other hand, in the case of the PtTFPP/PS sensor, the Stern–Volmer description is acceptable; the L or D models improve the flexibility in the response description without relevant inconsistencies, and the calibration results provided by the proposed PE model (with the evaluation dataset) are only slightly better than those provided by the corresponding classical model.

### 5.3. Response at Null Concentration and Sensitivity

The estimations of the response at null concentration and the sensitivity $K_{0}$ provided by each model at each calibration experiment are shown in Table 2. These parameters give a comparable physical interpretation for all the models used in the calibration. As can be observed, the estimations provided by the different calibration models are consistent (some differences are appreciated, particularly with $K_{0}$, associated with the fact that these estimations refer to the response at null concentration and correspond to an extrapolation of the calibration dataset). The consistency across models is better for $K_{1}$ and $K_{10}$ (i.e., estimated at 1 kPa and 10 kPa, respectively) because they are not affected by the extrapolation (see Appendix A).

### 5.4. Influence of the Instrumental Bias

In order to study the effect of an instrumental bias (either in the phase-shift or in the modulation-factor) over the accuracy provided by the calibration model, a sensitivity analysis has been performed. The plots in Figure 2 represent the RMS relative error as a function of the bias for the six experiments and for each calibration procedure (SV and L calibrations are omitted in experiments 5 and 6 because the results are out of the represented range). These plots represent the error for the evaluation dataset (similar plots are included in Appendix A for the calibration dataset). For negative bias in the instrument response (and even for a slightly positive bias in the case of the experiment 3), the results provided by L and D models degenerate to that of the SV model because, in the calibration procedure, the model parameters are constrained to values with physical sense (i.e., *x* in the rank [0,1] and positive values for the rest of parameters), and, when parameter *x* in the L or D models takes the value x=1, these models degenerate into a SV model. The fact that 5 models (SV, L, D, P2 and PE1) converge to the SV model for a very particular value of the instrumental bias is remarkable (for this bias, the optimal α in the PE1 model is –1, the $a_{2}$ coefficient of the P2 model is null, and the *x* parameter in the L and D models is equal to 1). The PE models provide a good robustness against instrumental bias, with the lowest relative error in experiments 1, 2, 5, and 6 and results very close to those of the best models in experiments 3 and 4.

The comparison of the results for the 3-parameter P2, L, and PE1 models reveals more robustness against instrumental bias and better performance with the PE1 model, suggesting that the inclusion of the exponent α in the model provides a useful flexibility in the description of the relationship between the analytical signal and the oxygen concentration.

### 5.5. Primary Measurements vs. Apparent Lifetimes

The comparison of the errors in Figure 2 or in Table 1 (B) shows that there are only slight differences between the results using the phase-shift as analytical signal and those using the phase-shift based lifetime. This is due to the relationship between both analytical signals (which are related by the tangent function, close to a linear relationship for small values of ϕ): since the phase-shift range is between 20 and 61 degrees in experiment 1, and between 29 and 53 degrees in experiment 5, the nonlinear effect of the transformation between ϕ and $\tau_{\phi }$ is almost irrelevant. These differences are more important when the modulation factor is transformed into the lifetime (experiments 3 and 4) because of the nonlinear relation between *m* and $\tau_{m}$.

According to the description of the photoluminescence provided in the literature [3,5,6,7], the calibration with the SV, L, and D models should be performed using either luminescence intensity (at null modulation frequency) or lifetime as analytical signal (or modulation factor if the modulation frequency can be considered very low), and the use of the modulation-factor or the phase-shift as analytical signal for the model calibration would be an inconsistency. However, in the case of phase-resolved photoluminescence with moderate modulation frequency, or if the measurements are affected by instrumental bias, additional inconsistencies are affecting the calibration model. In such situations, and due to the nonlinear monotonic relation between the modulation-factor or the phase-shift and their respective apparent lifetimes, the inconsistency associated with the use of *m* or ϕ (instead of $\tau_{m}$ or $\tau_{\phi }$) could be considered as acceptable as the use of a SV, L, or D models for a non-first order photoluminescence system operating with a moderate or high modulation frequency. This could explain the fact that, for some experiments and models (and at certain bias conditions), the results using the primary measurements (*m* or ϕ) are better than those using the corresponding apparent lifetimes ($\tau_{m}$ or $\tau_{\phi }$).

## 6. Conclusions

In this paper, we propose a new model for describing the relationship between the analytical signal and the analyte concentration in chemical sensors based on photoluminescence with modulated light sources. This model represents the concentration *C* as a *p*-th degree polynomial of $\phi^{\alpha }$, where ϕ is the analytical signal and α is an exponent to be fitted. The proposed model can be considered an extension of the Stern–Volmer model (it is identical to the Stern–Volmer model for p=1 and α=−1). As the Demas model, the proposed polynomial-exponent model can be scaled, by increasing the polynomial degree in order to provide a more detailed description of the relationship between the analytical signal and the analyte concentration. However, the procedure for fitting the model parameters from the calibration dataset is easier with the polynomial-exponent model than with the Demas model because, in the last one, calibrating a *N*-sites Demas model requires a *N*-dimensional search (of the *N* Stern–Volmer constants) for fitting the 2N parameters, while calibration of the polynomial-exponent model requires a one-dimensional, search of the exponent independently of the polynomial degree (since the polynomial coefficients are directly estimated in one step).

The proposed model (for degrees 1 and 2) has been compared with the classical calibration models (Stern–Volmer, Lehrer, Demas) in six experiments of phase-resolved photoluminescence. The proposed model provides the best trade-off between accuracy and complexity: the PE1 model (with only three parameters) provide an accuracy that is closer to the 4-parameter models than to the 3-parameter models. The PE2 model (with four parameters) provides better results than the 2-sites Demas model (also with four parameters).

Even though the parameters of the proposed model do not have a direct physical interpretation, the response of the model at null concentration and the sensitivity *K* can be derived from the model parameters. Consistent results are observed when these parameters estimated for the proposed model are compared to those estimated with the classical calibration models.

According to the analysis included in this manuscript, for photoluminescence sensors close to the Stern–Volmer model or excited at low modulation frequency, the Lehrer and Demas models are expected to be consistent. However, for higher modulation frequencies and particularly when the photoluminescence sensor significantly deviates from the first order differential equation, the Lehrer and Demas models exhibit inconsistencies and an alternative like the proposed polynomial-exponent model could be an interesting approach for calibrating the photoluminescence sensors.

## Figures and Tables

**Figure 1 sensors-20-04635-f001:**
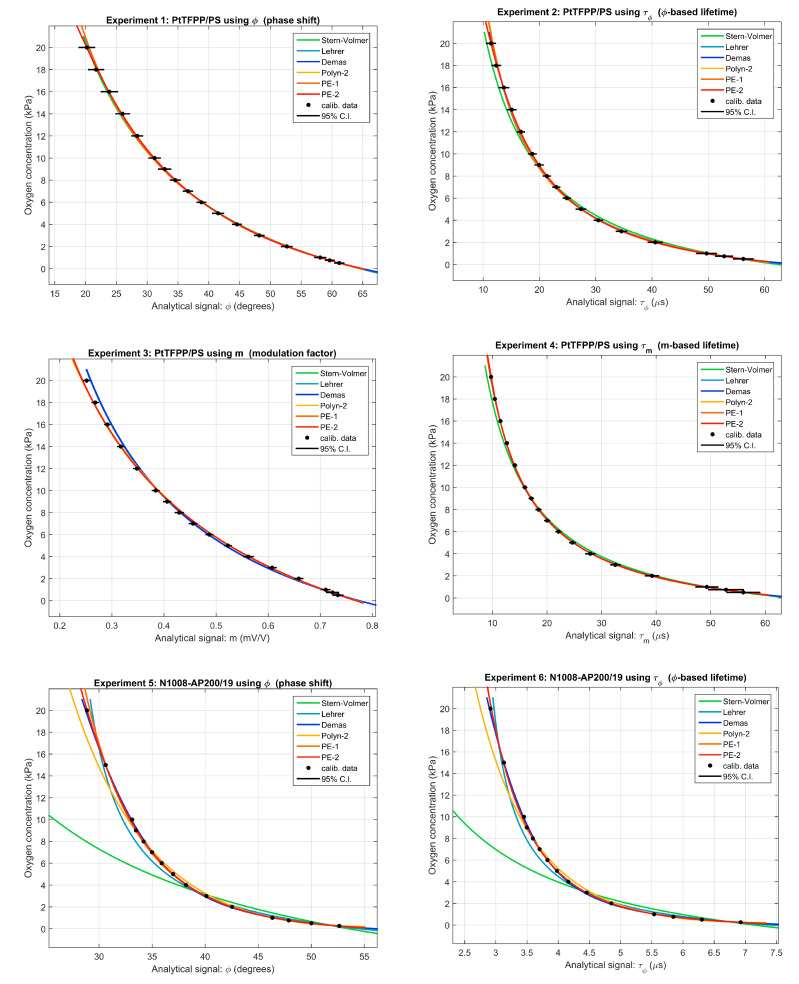
Calibration curves for the six experiments included in this study: oxygen concentration ($pO_{2}$, kPa) as a function of the analytical signal. The circles correspond to the calibration datas (the error lines represent the 95% confidence intervals of the calibration data); the curves are the calibration functions provided by the SV, L, D, P2, PE1, and PE2 models.

**Figure 2 sensors-20-04635-f002:**
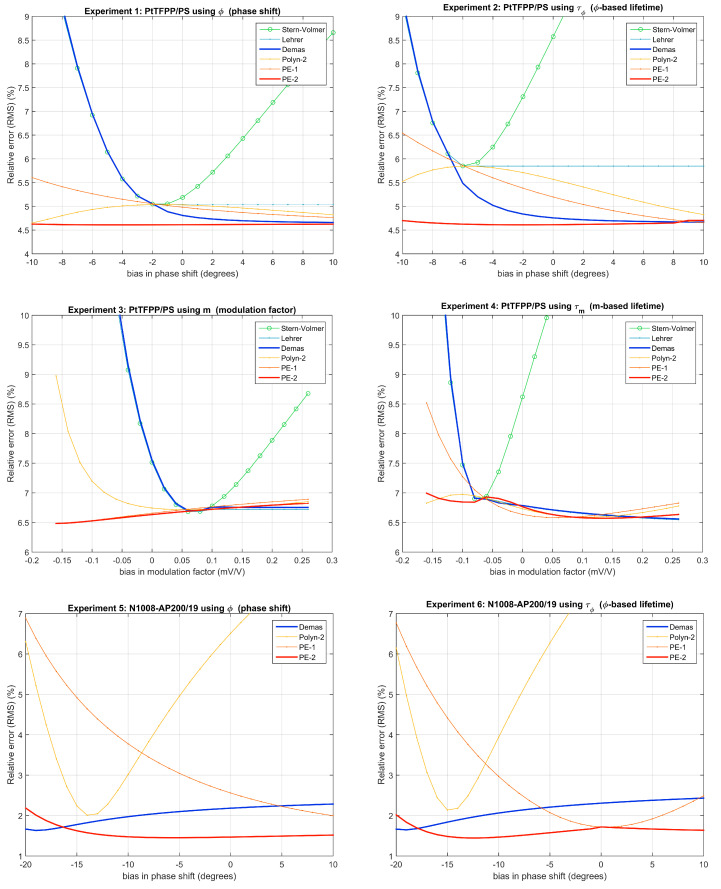
Effect of a bias in the instrumental measurement over the relative error in the evaluation datasets.

**Table 1 sensors-20-04635-t001:** Accuracy of the different calibration models for each experiment (the number of parameters to be fitted in each calibration model is indicated in parentheses).

(A) RMS Relative Error (%) for Average Calibration Data								
		**Experiment**						
**Model**		**1**	**2**	**3**	**4**	**5**	**6**	**Average**
SV	(2)	1.94	6.66	5.77	4.58	36.2	38.5	15.6
L	(3)	1.65	2.97	5.77	1.55	12.2	13.0	6.19
P2	(3)	1.64	2.56	4.12	1.56	6.48	8.10	4.08
PE1	(3)	1.56	1.97	3.70	1.84	2.54	1.66	2.21
D	(4)	1.34	1.25	5.79	1.55	2.17	2.30	2.40
PE2	(4)	0.79	0.86	3.64	1.51	1.41	1.66	1.65
**(B) RMS Relative Error (%) for Individual Evaluation Data **								
		**Experiment**						
**Model**		**1**	**2**	**3**	**4**	**5**	**6**	**Average**
SV	(2)	5.19	8.57	7.51	8.60	36.1	38.5	17.4
L	(3)	5.04	5.85	7.51	6.76	12.2	12.9	8.38
P2	(3)	5.03	5.57	6.74	6.71	6.51	8.13	6.45
PE1	(3)	4.98	5.20	6.66	6.61	2.56	1.72	4.62
D	(4)	4.81	4.76	7.55	6.76	2.18	2.31	4.73
PE2	(4)	4.61	4.62	6.63	6.74	1.47	1.72	4.30

**Table 2 sensors-20-04635-t002:** Response at null concentration and sensitivity at null concentration provided by each model at each calibration experiment.

	Response at Null Concentration						Sensitivity at Null Concentration $K_{0}$					
	**Experiment**						**Experiment**					
	**1**	**2**	**3**	**4**	**5**	**6**	**1**	**2**	**3**	**4**	**5**	**6**
**Model**	**(** ^*o*^ **)**	**(**μs**)**	**(mV/V)**	**(** μ **s)**	**(** ^*o*^ **)**	**(**μs**)**	**(kPa** ${}^{-1}$ **)**					
SV	64.5	62.5	0.775	64.5	53.5	7.14	0.109	0.244	0.100	0.308	0.108	0.198
L	64.6	64.0	0.775	66.1	54.6	7.41	0.112	0.285	0.100	0.351	0.178	0.343
P2	64.6	64.3	0.770	66.2	55.0	7.13	0.112	0.296	0.090	0.355	0.248	0.383
PE1	64.6	64.9	0.768	66.8	56.5	8.31	0.113	0.317	0.086	0.376	0.348	1.111
D	64.9	66.0	0.775	66.1	56.1	7.84	0.121	0.359	0.099	0.350	0.291	0.596
PE2	64.9	66.2	0.768	66.1	57.0	8.32	0.121	0.365	0.086	0.350	0.406	1.122

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
