# Peer review of "A Polynomial-Exponent Model for Calibrating the Frequency Response of Photoluminescence-Based Sensors"

_sensors, 2020, doi:10.3390/s20164635_

Round 1

Reviewer 1 Report

In "A polynomial-exponent model for calibrating the frequency response of photoluminescence-based sensors" authors illustrated a model to describe the relationship between oxygen concentration and the instrument response in case of photoluminescence sensors based on frequency-modulated LED. Unlike standard models, the "polynomial-exponential" model here described takes into consideration the presence of non-linear effect due to electronics, which is required to acquire and amplify the amplitude of signals at the modulation frequency. Results show that "PE" model offers benefits in terms of calibration errors especially when the modulation frequency is relative high (experiment number 5 and 6). In the reviewer opinion, the approach and results are surely interesting and promising even if some points should be remarked in order to fully understand the potentialities of the model and the possibility to generalize the results obtained in the manuscript. Here below it is reported a list of points that in the reviewer opinion should be discussed by the authors.

  • In order to validate the goodness and benefits of proposed model, in the reviewer opinion the calibration curves should be tested on an independent set of measurements (as mentioned by the authors themselves in the introduction) especially when models include a relative high number of parameters. The estimation of RMSE in test is of fundamental importance to avoid that model outperforms the other ones just because it includes the noise in the model. Since dataset comprises 25 measurements for each concentration, is it possible to split them into two datasets utilizing one for the calibration and the other for the test? This would rule out the presence of data overfitting.
  • The model proposed basically helps to compensate electronics non-linearities. In the reviewer opinion it would be helpful if the authors will include in the text a comment about the possibility to generalize this approach or if the model applicability is strictly related to the set-up/ instruments adopted in the manuscript.
  • May the authors provide additional information about measurements: are the different oxygen concentrations measured in a sequence or random order? The 25 measurements for each concentration were acquired consecutively, after some time or after having measured other concentrations? This point is very important to understand if systematic errors could be responsible for non linearities in the calibration curves.
  • May the authors provide any information about parameters robustness, e.g. the confidence interval of parameters obtained during the fitting?
  • Usually sensitivity is related to the whole concentration range of sensors, then it is strictly related to the working point (in this case the concentration). In the reviewer opinion it would be appropriate to utilize a notation that highlight this point (for example S0 or S(0) or S(C0)). Secondly sensitivity estimated with polynomial PE1 and PE2 are higher than those resulting by other fitting equations. Is in the authors’ opinion these values reliable?

Reviewer 2 Report

This manuscript discusses a calibration model for photoluminescence (PL) sensors that combines the use of polynomial and exponential functions to fit analytical response data vs the concentrations of the analyte. The PL solid state oxygen sensors exhibit heterogeneity and therefore classic calibration models have limitations. The proposed model is an upgrade to currently used calibration models. This model can be scaled to more fit data using more parameters. The described model shows a good trade-off between the number of fit parameters and accuracy of fitting the data.

The paper is composed well, with the clear introduction and discussion of limitations of current models. The polynomial-exponent model is discussed clearly, with details provided in the supplementary materials section. Authors designed experiments to evaluate the accuracy of the proposed model. PE model showed about 2% relative error, a significant improvement comparing to current models. 

I would like to suggest to use curve's colors that differ more in Fig.1, for better representation of calibration curves. 

Reviewer 3 Report

   This paper presents a model to calibrate photoluminiscent sensors. The approach is quite interesting, the data show that the authors proposal is the one with the smallest deviation. However, I am concerned about the measurements devition of the data showed in Figure 1. The authors said that each data point is the average of 25 measurements, however, they do not show the dispersion or the standard deviation of each point in order to support the assertion that their model is better than SV, L or D models. Furthermore, what is the meaninig of the obtained small error in terms of concentration? If the concentration data dispersion is larger than the models errors, then it can be necessary a further discussion about how much is their model better than the other ones.

   On the other hand, in page 13, first paragraph, line 8, the word "model" is repeated two times. Another spelling revision of the manuscript must be performed.

Round 2

Reviewer 1 Report

Authors successfully commented reviewer remarks and improved the manuscript that, according to review opinion, is now worth to be published in the present form.